# Upregulation of the Renin–Angiotensin System Is Associated with Patient Survival and the Tumour Microenvironment in Glioblastoma

**DOI:** 10.3390/cells13070634

**Published:** 2024-04-05

**Authors:** Mathew Lozinski, Eugenie R. Lumbers, Nikola A. Bowden, Jennifer H. Martin, Michael F. Fay, Kirsty G. Pringle, Paul A. Tooney

**Affiliations:** 1School of Medicine and Public Health, College of Health, Medicine and Wellbeing, University of Newcastle, Callaghan, NSW 2308, Australia; mathew.lozinski@newcastle.edu.au (M.L.); nikola.bowden@newcastle.edu.au (N.A.B.); jenniferh.martin@newcastle.edu.au (J.H.M.); michael.fay@newcastle.edu.au (M.F.F.); 2Mark Hughes Foundation Centre for Brain Cancer Research, University of Newcastle, Callaghan, NSW 2308, Australia; 3Drug Repurposing and Medicines Research Program, Hunter Medical Research Institute, New Lambton Heights, NSW 2305, Australia; 4School of Biomedical Sciences and Pharmacy, College of Health, Medicine and Wellbeing, University of Newcastle, Callaghan, NSW 2308, Australia; eugenie.lumbers@newcastle.edu.au (E.R.L.); kirsty.pringle@newcastle.edu.au (K.G.P.); 5Mothers and Babies Research Program, Hunter Medical Research Institute, New Lambton Heights, NSW 2305, Australia; 6GenesisCare, Gateshead, NSW 2290, Australia

**Keywords:** glioblastoma, prorenin, angiotensinogen, hypoxia, chemoradiation

## Abstract

Glioblastoma is a highly aggressive disease with poor survival outcomes. An emerging body of literature links the role of the renin–angiotensin system (RAS), well-known for its function in the cardiovascular system, to the progression of cancers. We studied the expression of RAS-related genes (*ATP6AP2*, *AGTR1*, *AGTR2*, *ACE*, *AGT*, and *REN*) in The Cancer Genome Atlas (TCGA) glioblastoma cohort, their relationship to patient survival, and association with tumour microenvironment pathways. The expression of RAS genes was then examined in 12 patient-derived glioblastoma cell lines treated with chemoradiation. In cases of glioblastoma within the TCGA, *ATP6AP2*, *AGTR1*, *ACE*, and *AGT* had consistent expressions across samples, while *AGTR2* and *REN* were lowly expressed. High expression of *AGTR1* was independently associated with lower progression-free survival (PFS) (*p* = 0.01) and had a non-significant trend for overall survival (OS) after multivariate analysis (*p* = 0.095). The combined expression of RAS receptors (*ATP6AP2*, *AGTR1*, and *AGTR2*) was positively associated with gene pathways involved in hypoxia, microvasculature, stem cell plasticity, and the molecular characterisation of glioblastoma subtypes. In patient-derived glioblastoma cell lines, *ATP6AP2* and *AGTR1* were upregulated after chemoradiotherapy and correlated with an increase in *HIF1A* expression. This data suggests the RAS is correlated with changes in the tumour microenvironment and associated with glioblastoma survival outcomes.

## 1. Introduction

Glioblastoma is the most prevalent and aggressive primary brain tumour in adults. Despite decades of research, advances in treatment over the past twenty years have been minimal. Glioblastoma patients face a grim prognosis, including a median survival time of 9–10 months and a five-year survival rate less than 5% [1,2,3]. An aggressive multi-disciplinary treatment regimen is employed for newly diagnosed glioblastoma, involving maximal safe surgical resection followed by concomitant radiation therapy (RT) with temozolomide (TMZ) chemotherapy and subsequent TMZ maintenance treatment [4]. Despite this, progression at the primary tumour site often occurs and presents a formidable challenge in the treatment of glioblastoma. The exceptional aggressiveness of glioblastoma can be attributed to a multitude of factors that foster resistance to TMZ and RT, including the presence of glioma stem cells (GSCs) [5]; an enhanced capacity to repair DNA damage induced by chemoradiation [6], tumour vasculature, hypoxic regions; and a heterogenous and immunosuppressive tumour microenvironment (TME) [7,8,9,10].

The circulating renin–angiotensinogen system (RAS) plays a crucial role in regulating blood pressure, fluid balance and homeostasis in humans [11]. This system involves a multi-step process that produces the main effector hormone, angiotensin II (Ang II), which induces these physiological changes. Angiotensinogen, encoded by the *AGT* gene, is primarily produced in the liver and is converted to angiotensin I (Ang I) when cleaved by renin. Active renin is only produced in the juxtaglomerular apparatus of the kidneys. Other tissues produce the precursor prorenin (encoded by the *REN* gene), which is inactive because it has a 28 amino acid pro-segment that must be unfolded out of the active site or removed by enzymes [12]. The pro-segment unfolds when prorenin binds to the prorenin receptor (PRR; encoded by *ATP6AP2*). Other proteolytic enzymes besides renin, e.g., cathepsin, can also form angiotensin peptides [13]. Ang I formed from the interaction of renin with angiotensinogen (AGT) is converted to Ang II by the angiotensin-converting enzyme (ACE), located predominantly in the lungs. Downstream physiological changes occur after the binding of Ang II to angiotensin II receptor type 1 (AGTR1) and angiotensin II type 2 receptor (AGTR2). Although the RAS was originally considered a systemic/endocrine system, RAS components are known to be expressed locally in many tissues including the brain [14]. For instance, angiotensinogen is produced, secreted, and intracellularly retained by astrocytes and neurons where it is also converted to various neuropeptides that induce receptor signalling [14,15]. Moreover, an increasing body of literature suggests that the RAS is associated with a number of hallmarks of cancer and plays a role in promoting tumour growth and the development of new blood vessels through angiogenesis [16,17].

In glioblastoma, the expression of RAS components may facilitate tumour growth and be associate with patient survival [18,19]. The expression of RAS receptors (AGTR1, AGTR2, and ATP6AP2) has been demonstrated on the glioblastoma microvasculature [20,21], and the inhibition of AGTR1 or AGTR2 reduced glioblastoma cell growth in vitro and in vivo [19,22,23]. Furthermore, low expression of AGT has been associated with a favourable response to the anti-angiogenic treatment, bevacizumab, in glioblastoma patients at progression [24,25]. This suggests a potential role for the RAS in the growth of glioblastoma cells and a potential association with the TME.

Most studies have focused on specific RAS components and their relationship to glioblastoma growth; however, the link between the RAS and TME factors specific for glioblastoma such as stemness and hypoxia has not been fully explored. Where a number of RAS genes/proteins were explored in great depth, only small numbers of samples were studied [20,26]. Furthermore, the expression of the RAS after standard chemoradiation treatment has not been explored in glioblastoma cells. Here, we investigated the gene expression of RAS components (*ATP6AP2*, *AGTR1*, *AGTR2*, *ACE*, *AGT*, and *REN*) in glioblastoma patient samples from The Cancer Genome Atlas (TCGA) and their association with survival outcomes and the expression of TME pathways. Additionally, we observed expression changes in RAS genes after chemoradiation treatment in 12 patient-derived glioblastoma cell lines.

## 2. Materials and Methods

### 2.1. Ethics

This study was approved by the Human Research Ethics Committee of the University of Newcastle (H-2020-0389).

### 2.2. TCGA In Silico Analysis

To study the expression of RAS genes (*ATP6AP2*, *AGTR1*, *AGTR2*, *ACE*, *AGT*, and *REN*) in newly diagnosed glioblastoma, RNA-seq via expected maximisation (RSEM) data were collated from the TCGA PanCancer Atlas study [27,28] (downloaded on 17 August 2023). Survival analysis was performed on glioblastoma data (*n* = 144) within the PanCancer Atlas study where progression-free survival (PFS) and overall survival (OS) data were present. TCGA cases were stratified into “high” and “low” expressions based on a median split of RSEM expression values per gene. The log-rank test was used to find potential differences in overall survival (OS) between groups. Multiple Cox regression was used on genes with a significant difference in OS from the log-rank test (*p* < 0.05), using clinical covariables that were determined to be significant through univariate Cox regression (Appendix A).

Single-sample gene set enrichment analysis (ssGSEA) was performed using GenePattern [29] on glioblastomas (*n* = 170) and low-grade gliomas (LGGs) (*n* = 534) RNA-seq data (transcripts per million; TPM) downloaded from the Genomics Data Commons (GDC) Data Portal (https://portal.gdc.cancer.gov/, downloaded on 17 August 2023). Enrichment scores represented the degree to which the gene sets were up- or downregulated within a sample [30]. The RAS receptor gene set consisted of *ATP6AP2*, *AGTR1*, and *AGTR2*. TME-related gene sets were chosen for analysis, including hypoxia activation, mature vasculature, microvasculature [31], glioma stem cell plasticity [32], and the classical, mesenchymal, and proneural subtypes [33] (Appendix A).

### 2.3. Cell Lines and Reagents

Twelve patient-derived glioblastoma cell lines (BAH1, FPW1, JK2, MMK1, RKI1, HW1, PB1, SB2b, SJH1, RN1, MN1, and WK1) were kindly provided by Professor Bryan Day (QIMR Berghofer Medical Research Institute, Brisbane, Australia). The cell lines are fully characterised with publicly available molecular and patient data, published by Stringer et al. [34]. Cells were grown as adherent monolayers in Matrigel^®^ (Corning^®^, Corning, NY, USA)-coated tissue culture flasks in serum-free conditions, as previously described [6]. TMZ was purchased from Sigma-Aldrich (St. Louis, MO, USA), aliquoted in dimethyl sulfoxide (DMSO) (100 mM), and stored between 2–8 °C. RT was delivered using an RS-2000 Small Animal Irradiator (Rad Source, Buford, GA, USA) [35].

### 2.4. RNA Sequencing Analysis

RNA extraction and whole transcriptomic sequencing was performed on glioblastoma cell lines (*n* = 12) that were treated with DMSO control or TMZ (35 μM) + RT (2 Gy), 4 days post-treatment, as previously described [35]. This timepoint was chosen as it represents the time at which we first observed the effect of TMZ + RT on glioblastoma cell growth [35]. TPM expression values were used for comparison between TMZ + RT treatment and untreated (DMSO control) groups. Using TPM values, gene expression changes (Δ_expression_) were calculated by dividing the TPM of the TMZ + RT-treated sample by the expression value of the DMSO control. RAS gene Δ_expression_ values were correlated with *HIF1A* Δ_expression_ to identify whether the up- or downregulation of each individual RAS gene was associated with *HIF1A* expression change after TMZ + RT treatment.

### 2.5. Statistical Analysis

The statistical analyses including Student’s *t*-test for comparison between two groups and correlation analysis using the Pearson correlation, all performed in GraphPad Prism 10. Unsupervised hierarchical clustering (Ward’s method) of ssGSEA scores was performed in R using the ‘stats’ package and visualised using the ‘ComplexHeatmap’ package. Survival analyses using the log-rank test and Cox regression were performed in R using the ‘survival’ package. *p*-values < 0.05 were considered significant.

## 3. Results

### 3.1. Higher Gene Expression of RAS Components Associates with Poorer Survival Outcomes for Glioblastoma

We first investigated the gene expression of RAS components within the publicly available TCGA PanCancer database [36]. The expression profile within glioblastoma cases varied across the different RAS components (Figure 1A). The majority of glioblastoma cases expressed *ATP6AP2*, *AGTR1*, *ACE*, and *AGT* but not *AGTR2* and *REN* (Figure 1A). When compared to LGG cases, *ATP6AP2*, *AGTR1*, and *ACE* were significantly higher in expression in glioblastomas, with *AGTR1* showing the highest change in expression (5.7-fold higher) (Figure 1B–E). Although *AGTR2* was significantly higher in glioblastomas compared to LGG cases, this gene was not expressed in most glioblastoma cases (Figure 1D). Conversely, *AGT* and *REN* had lower expressions in glioblastomas compared to low-grade gliomas (Figure 1F,G). The level of expression of *ATP6AP2*, *AGTR1*, *AGTR2*, and *ACE* genes in glioblastomas was similar to that observed in a range of other solid tumours and haematological cancers (Appendix A). Interestingly, *AGT* was higher in expression in both glioblastomas and low-grade gliomas, while *REN* expression was comparatively lower when compared to the majority of the other tumour types (Appendix A).

In addition to the six RAS genes, the expressions of *ACE2* and *MAS1* were investigated (Appendix A). ACE2 and MAS1 are known to oppose the physiological effects of elevated RAS activity [37,38]. Both *ACE2* and *MAS1* were significantly lower in expression in glioblastomas compared to low-grade gliomas and were decreased in glioblastomas compared to other tumour types (Appendix A).

Progression-free survival (PFS) and overall survival (OS) were compared across TCGA glioblastoma cases, stratified from high and low expression of *ATP6AP2*, *AGTR1*, *AGTR2*, *ACE, AGT*, and *REN* genes. Univariate Cox regression analysis showed high *AGTR1* and *ATP6AP2* expressions associated with lower PFS (Appendix A). After correction for significant clinical features (therapy type and isocitrate dehydrogenase (IDH) mutation status), *AGTR1* remained significant (HR: 1.77 (1.15–2.73), *p* = 0.01) while *ATP6AP2* had a non-significant trend (HR: 1.49 (0.99–2.23), *p* = 0.055) (Figure 2A,B, Appendix A). *AGTR2*, *ACE*, *AGT*, and *REN* were not significantly associated with PFS. With respect to OS, high *AGTR1* expression was significantly associated with lower OS in a univariate Cox analysis (Appendix A); however, after multivariate analysis, *AGTR1* was non-significantly associated with OS (HR = 1.55 (0.93–2.59), *p* = 0.095) (Figure 2C, Appendix A). Although the usual method for survival analysis requires variables to be significant in univariate analysis prior to multivariate analysis, there are instances of multivariate analysis revealing significance despite non-significance through univariate Cox regression [39]. Interestingly, high *REN* expression was significantly associated with poorer OS in glioblastoma cases after multivariate analysis (HR = 2.25 (1.05–4.8), *p* = 0.036), despite a non-significant trend after univariate Cox regression (HR = 1.53 (0.89–2.65), *p* = 0.13) (Figure 2D, Appendix A). This could be due to the low number of cases in the high *REN* expression group (*n* = 18) as the majority of samples had no detectable gene expression of *REN* (*n* = 126).

To further explore the genomic data of the TCGA PanCancer dataset, copy number alterations (CNAs) of RAS genes were investigated. *ATP6AP2*, *AGTR1*, *AGTR2*, *AGT*, and *ACE* had low frequencies of CNAs (<1%) in glioblastomas; however, *REN* was altered in 6% of glioblastoma cases (Appendix A). The majority of CNAs of *REN* were amplifications across all PanCancer tumour types, where glioblastoma was ranked the third most common (Appendix A). Intriguingly, *REN* copy number alterations across all cancers were associated with lower PFS and OS when corrected for age, sex, and tumour type (Appendix A), suggesting *REN* may play a wider role in cancer progression. In glioblastomas, however, a non-significant trend was observed for PFS (HR = 1.33 (0.94–1.9), *p* = 0.11) and OS (HR = 1.37 (0.94–1.98), *p* = 0.1) where the *REN* altered group (*n* = 36) had shorter PFS (6.7 months) and OS (11.1 months) compared to the non-altered group (*n* = 542) (PFS: 7.2 months, OS: 14.5 months) (Appendix A).

### 3.2. RAS Receptor Expression Correlates with Tumour Microenvironment Features in Glioblastoma

To explore the relationship of the RAS pathway with the tumour microenvironment, ssGSEA was performed on the gene expression of RAS receptors (*ATP6AP2*, *AGTR1*, and *AGTR2*) and microenvironment pathways such as hypoxia, mature vasculature/microvasculature, stem cell plasticity, as well as the established molecular subtypes of glioblastoma (i.e., classical, mesenchymal, and proneural) [33]. As RAS receptors play a direct role in the production or actions of Ang II and the subsequent physiological changes associated with this pathway, these genes were examined. In glioblastoma samples (*n* = 170), RAS receptor expression was positively correlated with genes involved in hypoxia activation (r = 0.35, *p* < 0.0001), microvasculature (r = 0.45, *p* < 0.0001), stem cell plasticity (r = 0.20, *p* = 0.009), as well as the classical (r = −0.38, *p* < 0.0001) and mesenchymal glioblastoma subtypes (r = 0.44, *p* < 0.0001) and negatively correlated with the proneural subtype (r = −0.38, *p* < 0.0001) (Figure 3). This trend was also observed in the total glioma cases (combined glioblastoma and LGG) (*n* = 704), while RAS receptor expression was significantly increased in glioblastoma cases compared to LGG (*p* < 0.0001) (Appendix A). This suggests that glioblastoma tumours with higher RAS receptor expression also display higher hypoxia, microvasculature, and stem cell expression, as well as being associated with the more aggressive mesenchymal subtype while having lower expression of the treatment sensitive proneural subtype.

### 3.3. Glioblastoma Chemoradiation Increases Gene Expression of RAS Components in Patient-Derived Cell Lines

The baseline mRNA expression of RAS genes (*ATP6AP2*, *AGTR1*, *AGTR2*, *AGT*, *ACE*, and *REN*) were explored in 12 patient-derived glioblastoma cell lines (Figure 4A). A similar expression profile in the cell lines compared to the TCGA data (Figure 1A) was evident, suggesting glioblastoma cells retain the expression of RAS genes in vitro. The expression of RAS genes was investigated after the combination of a clinically relevant dose of TMZ (35 μM) with RT (2 Gy) (Figure 4B–H). Previously, we observed a significant increase in expression of DNA repair and pro-inflammatory-related genes after chemoradiation across the 12 glioblastoma cell lines [35]. The combination of TMZ + RT resulted in a slight but significant increase in the expression of *ATP6AP2*, *AGTR1*, and *ACE*, while *AGT* and the lowly expressed *AGTR2* and *REN* remained unchanged across all 12 glioblastoma cell lines (Figure 4B–G). HIF1A plays a crucial role in the cellular response to hypoxic conditions [40]. As hypoxia activation was positively correlated with RAS receptor expression (Figure 3B), the association with *HIF1A* and *RAS* gene expression was investigated after glioblastoma cell lines were treated with chemoradiation. This gene was significantly upregulated across glioblastoma cell lines after TMZ + RT treatment (Figure 4H). When comparing the change of expression caused by chemoradiation (TMZ + RT), *ATP6AP2* (r = 0.53, *p* = 0.007), *AGTR1* (r = 0.43, *p* = 0.043), and *AGT* (r = 0.55, *p* = 0.006) were positively correlated with *HIF1A* expression changes, while *REN*, *AGTR2* and *ACE* had no significant correlation. This data suggests that chemoradiation may stimulate the expression of RAS components in a manner that associates with an increase in hypoxia expression in glioblastoma cells.

## 4. Discussion

Glioblastoma is an exceptionally aggressive and invasive brain cancer, often prone to post-treatment resistance and progression resulting in poor patient survival. There is an emerging role for the renin–angiotensin system in tumour progression [17], which may play a role in glioblastoma and facilitate a pro-tumour microenvironment [41]. Here, we investigated the expression of six key RAS genes and their relationship to patient survival, microenvironmental phenotypes, and response to standard chemoradiation.

Firstly, we observed the expression of RAS genes in a TCGA PanCancer cohort. The majority of genes (*ATP6AP2*, *AGTR1*, *ACE*, and *AGT*) were consistently expressed in glioblastoma cases, with the exception of *AGTR2* and *REN*. *REN*, which encodes the prorenin protein, was poorly expressed across the glioblastoma samples and, overall, had lower expression in glioblastomas compared to LGGs. Juillerat-Jeanneret et al. [26] demonstrated relatively low expression of REN in a small sample (*n* = 3) of glioblastoma cases; it was mostly present on neurons and macrophages within the microenvironment and heterogeneously expressed in glioblastoma cells [26]. Overall, *REN* was lowly expressed in gliomas within the TCGA compared with the majority of other cancers, suggesting the source of prorenin for the activity of RAS in glioblastoma may be external to the tumour. Alternative pathways may also be a source of angiotensin, for instance cathepsin and tonin are known to be present in high quantities within the brain and can generate angiotensin peptides through AGT [42].

The majority of RAS genes (*ATP6AP2*, *AGTR1*, *AGTR2*, and *ACE*) in our study were highly expressed in glioblastomas compared to LGGs. As *ATP6AP2*, *AGTR1*, and *AGTR2* are the main receptors in the RAS; this suggests that glioblastomas have a greater capacity to utilise this pathway for their survival. Despite *AGT* having slightly lower expression in glioblastomas compared to LGGs, the gene was highly expressed in gliomas compared to other cancers in the TCGA with comparable expression in cancers of the liver (cholangiocarcinoma and hepatocellular carcinoma), the organ well described to be the main source of angiotensinogen [43]. Angiotensinogen is known to be expressed within the brain and highly expressed in glioblastoma patient samples [26,44]. Furthermore, low expression or promoter methylation of *AGT* has been associated with better response to bevacizumab at progression [24,25], suggesting its potential role in providing an alternative pathway for angiogenesis despite inhibition of vascular endothelial growth factor (VEGF).

Next, we evaluated the survival outcomes (PFS and OS) of glioblastoma cases based on the expression of each individual RAS gene. A non-significant trend was demonstrated for *ATP6AP2* where higher expression was associated with lower PFS, while *REN* was not significantly associated with OS in a univariate analysis, although it became significant with multivariate analysis. *AGTR1* was the only gene that was independently associated with a survival outcome, where higher expression correlated with shorter PFS in both univariate and multivariate analysis. *AGTR1* was significantly associated with OS in a univariate analysis; however, this was not significant after correction for clinical features in a multivariate Cox regression. *AGTR1* has been recently reported to be significantly associated with poorer PFS and/or OS in a TCGA glioblastoma cohort [19,22]; however, the authors did not perform multivariate analysis. Nevertheless, our data suggest that higher expression of RAS components contributes to poorer survival outcomes for patients with glioblastoma. Whilst one would expect that faster tumour growth requires better blood supply, the involvement of the RAS in glioblastoma requires further investigation.

Important drivers of glioblastoma progression are the aspects of the microenvironment that tumour cells reside in, such as hypoxia, microvasculature, and stem cell populations. We, thus, examined the relationship of RAS receptors (*ATP6AP2*, *AGTR1*, and *AGTR2*) with TME expression within TCGA glioblastoma samples. Unsurprisingly, RAS receptor expression was higher in glioblastomas compared to LGGs. There was a positive correlation with microvasculature and stem cell plasticity, which largely supports other studies that have linked RAS receptors with angiogenesis, and expression on microvessels and cancer stem cells in glioblastomas [19,20]. Glioblastoma tumours are known to have a highly hypoxic environment, facilitating treatment resistance and proliferation [45]. The link between the RAS and hypoxia has been explored in normal human cell lines [46,47,48]; however, this has not been directly investigated in glioblastoma cases. Our data demonstrate that increased RAS receptor expression was significantly correlated with increased expression of genes involved in hypoxia. Furthermore, tumours displaying a more mesenchymal phenotype had greater RAS receptor expression. The mesenchymal subtype carries the worst patient outcome, whereby it is associated with increased markers of angiogenesis and necrosis whilst also being most resistant to standard glioblastoma treatment [7,49]. ATP6AP2 has also been linked to the Wnt/β-catenin signalling pathway [21], which is associated with the proliferation of GSCs, treatment resistance, and the mesenchymal phenotype [50]. Overall, this suggests that glioblastomas with higher RAS receptor expression are associated with a more aggressive microenvironment. In future work, the link between the RAS and other TME pathways that influence a pro-tumour environment, such as the immune system and tumour heterogeneity [51,52], could also be explored.

Currently the RAS has not been investigated in the context of standard glioblastoma treatment and whether it may play a role in resistance to TMZ and/or RT. We used patient-derived glioblastoma cell lines which were grown in serum-free conditions and resemble the genomic and transcriptomic characteristics of the original patient tumours [34]. Baseline expressions of RAS genes were relatively similar to the TCGA glioblastoma cohort, where *REN* and *AGTR2* were very lowly expressed while *ATP6AP2*, *AGTR1*, *ACE,* and *AGT* were consistently expressed. When exposed to a clinically relevant dose of TMZ and RT, some RAS genes, including *ATP6AP2*, *AGTR1*, and *ACE*, showed a slight but significant increase in expression across the 12 cell lines. Interestingly, these expression changes were positively correlated with *HIF1A* expression changes, indicating a potential link between RAS expression and hypoxia induction in response to chemoradiation therapy.

Although this study did not explore the targeting of RAS components, inhibition of RAS receptors such as AGTR1, AGTR2, and ATP6AP2 have been demonstrated to reduce glioblastoma cell growth, proliferation, and/or angiogenesis [19,21,22,23,53,54]. Targeting renin using synthetic renin inhibitors induced apoptosis and reduced the growth of commercial glioma cells in vitro [26], while a similar observation was reported using an siRNA against ATP6AP2 [21]. A monoclonal antibody against ATP6AP2 was also shown to reduce glioma sphere growth, induce apoptosis and reduce tumour growth in a subcutaneous xenograft glioma model [53]. Losartan, an FDA-approved AGTR1 inhibitor, has been particularly explored in recent studies where it was shown to reduce glioma cell line growth in vitro as well as in vivo in a U87 xenograft model [22]. More recently, losartan was shown to induce favourable TME changes such as reduced angiogenesis, immunosuppression, and hypoxia-associated gene expression in a GL261 mouse model, increasing the survival of mice when treated with an immune checkpoint inhibitor [55]. Despite this, it is suggested that losartan is not blood–brain barrier permeable [56]. Furthermore, inhibition of AGTR2 using a derivative of EMA401, a preclinical drug investigated for peripheral neuropathic pain, reduced glioblastoma cell growth in vitro and in an intracranial mouse model [23]. A phase I, open-label, proof-of-concept trial of RAS modulators in recurrent glioblastomas showed promising results, where the median OS of patients was 19.9 months—higher than the 12–14 month median OS of glioblastoma patients in clinical trials but not statistically significant, perhaps due to the small sample size (*n* = 17) [57]. Overall, targeting the RAS is becoming a promising therapeutic strategy against glioblastoma and further work is needed to identify the most suitable drugs and whether their combination with existing treatments such as TMZ and RT are effective.

A limitation to our study is the transcriptional perspective from which we have examined RAS expression in glioblastoma cases from the TCGA and in patient-derived cell lines. As a result, this does not completely capture the scope of RAS expression. For instance, we showed a minority of TCGA glioblastoma samples had an expression of *AGTR2* (19/160), which aligned with a previous study that showed a lack of AGTR2 mRNA (nanostring expression) and protein expression (immunoblot) in a small cohort of glioblastoma cases; however, this protein was expressed on the microvessels and stem cells of glioblastoma through immunohistochemistry [20]. Furthermore, the single timepoint chosen to examine the gene expression of RAS in glioblastoma cells after chemoradiation (4 days post TMZ + RT) is a limitation as expression changes are likely to be transient across time. To examine the RAS and its relationship to TME factors more closely, future work could consider glioblastoma cells under hypoxic conditions, the use of organoid systems, as well as in vivo models where both the RAS transcriptome and proteome can be monitored. In future work, the use of siRNA or shRNA to knockdown such RAS genes in combination with chemoradiation may also give insight into the feasibility of targeting the RAS as well as to validate the observations of increased RAS gene expression in the glioblastoma cell lines.

In conclusion, this study demonstrates a connection between the expression of RAS components and survival outcomes in ex vivo glioblastoma cases. It strengthens the link between RAS and the TME in glioblastomas, where higher RAS expression is associated with a microvasculature, hypoxia, and mesenchymal expression signature. Furthermore, components of the RAS were observed to be influenced by glioblastoma standard treatment and future work is needed to identify the extent to which the RAS is involved in treatment resistance.

## Figures and Tables

**Figure 1 cells-13-00634-f001:**
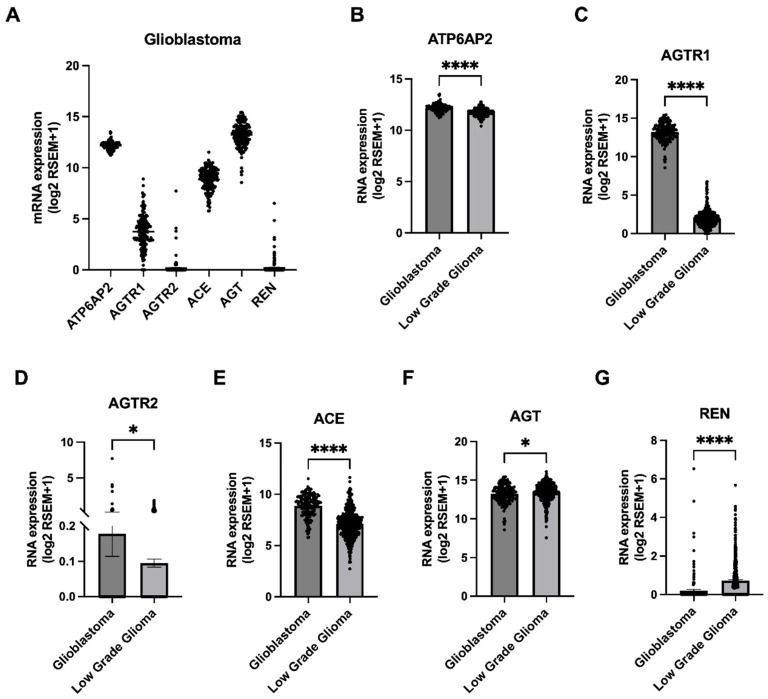
Expression of RAS genes in glioblastoma and low-grade glioma TCGA cases. (**A**) RNA expressions (log2 RSEM +1) of *ATP6AP2*, *AGTR1*, *AGTR2*, *ACE*, *AGT*, and *REN* genes are shown across TCGA PanCancer glioblastoma cases (*n* = 160). Levels of expression (mean ± SEM) were compared between TCGA PanCancer glioblastoma (*n* = 160) and LGG cases (*n* = 514) for *ATP6AP2* (**B**), *AGTR1* (**C**), *AGTR2* (**D**), *ACE* (**E**), *AGT* (**F**), and *REN* (**G**) genes using Student’s *t*-test (* *p* < 0.05 and **** *p* < 0.0001). See Appendix A for comparison of RAS gene expression across all PanCancer cancers.

**Figure 2 cells-13-00634-f002:**
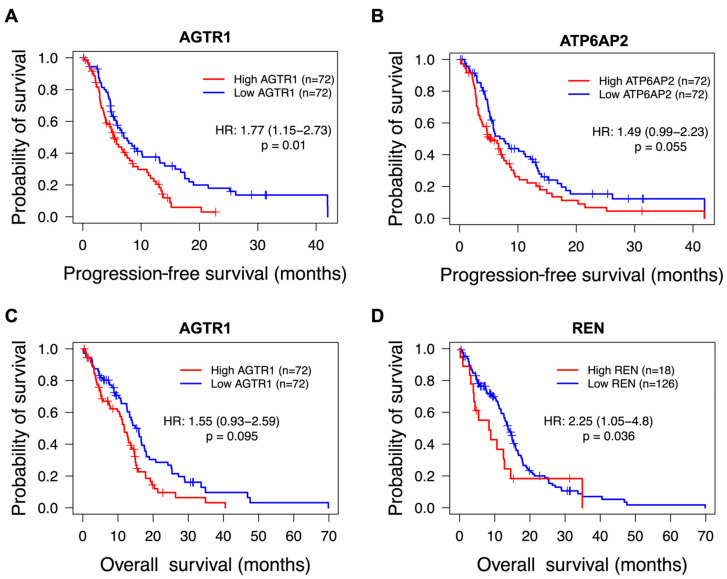
Survival plots of TCGA glioblastoma cases based on RAS gene expression. Glioblastoma cases (*n* = 144) were stratified into “high” (red) and “low” (blue) expression groups based on a median split of RNA expression for each respective RAS gene. A univariate log-rank test was performed for PFS and OS, followed by multivariate Cox regression to account for significant clinical factors (see Appendix A), with independently associated genes identified with *p* < 0.05 from both statistical tests. Hazard ratios and *p*-values are depicted from the multivariate Cox regression model. *AGTR1* was the only RAS gene independently associated with a survival outcome (**A**), where high expression was associated with poorer PFS. (**B**) High *ATP6AP2* was associated with poorer PFS in a log-rank test but was non-significant after multivariate Cox regression. (**C**) High *AGTR1* was associated with OS in a univariate analysis but had a non-significant trend after multivariate analysis. (**D**) High *REN* expression was not significant in a univariate analysis; but, after multivariate analysis, it was associated with lower OS. No other RAS gene (i.e., *AGTR2*, *AGT*, or *ACE*) showed significant differences in univariate or multivariate analysis.

**Figure 3 cells-13-00634-f003:**
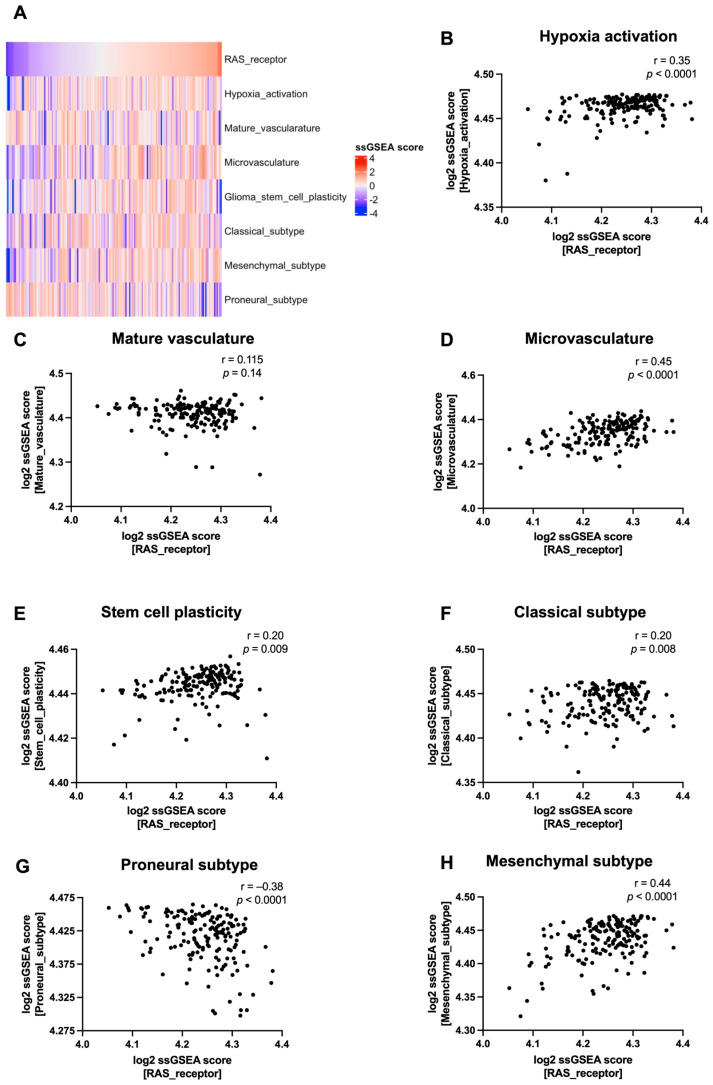
Relationship of RAS receptor expression to tumour microenvironment pathway expression in TCGA glioblastoma cases. (**A**) Log-transformed ssGSEA z-scores are represented for the RAS receptor pathway and TME-related pathways including hypoxia activation, mature vasculature, microvasculature, glioma stem cell plasticity, and the classical, proneural, and mesenchymal subtypes. (**B**–**H**) Pearson correlation was utilised to compare the ssGSEA score of the RAS receptor gene set with TME-related pathways (*p* < 0.05 was considered significant).

**Figure 4 cells-13-00634-f004:**
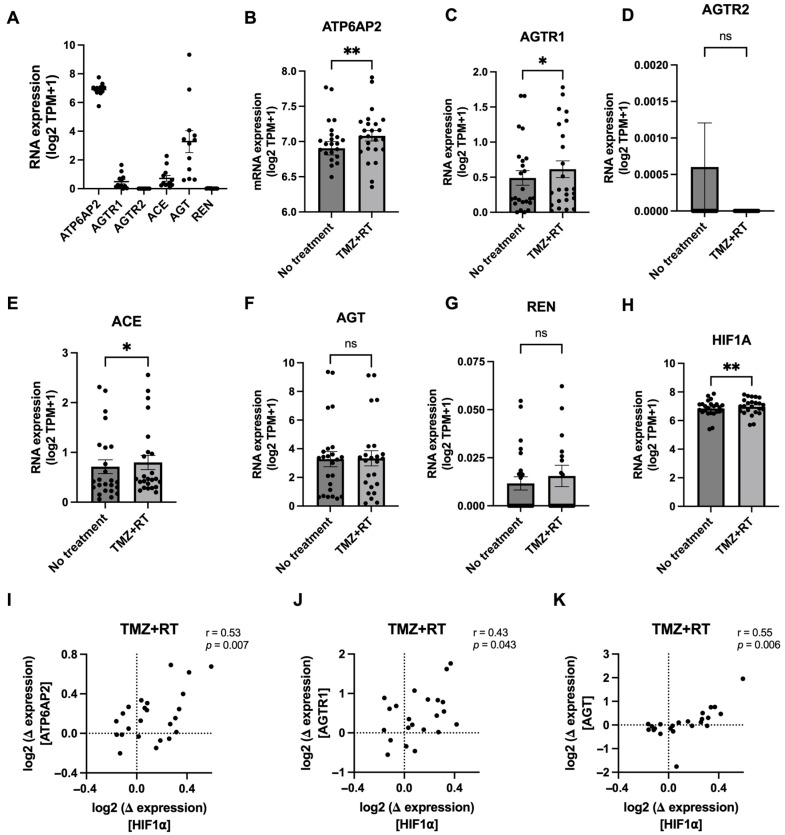
Expression of RAS genes after chemoradiation within patient-derived glioblastoma cell lines (*n* = 12). (**A**) Baseline RNA-seq expression (log2 TPM + 1) of RAS genes within the 12 patient-derived glioblastoma cell lines. (**B**–**H**) Gene expression of *ATP6AP2* (**B**), *AGTR1* (**C**), *AGTR2* (**D**), *ACE* (**E**), *AGT* (**F**), *REN* (**G**), and *HIF1A* (**H**) after standard treatment with a clinically relevant dose of TMZ (35 μM) and RT (2 Gy), 4 days post-treatment. Data points (*n* = 24) represent each biological replicate per cell line with bar plots representing the mean ± SEM. A paired t-test was used to perform statistical analysis (*p* < 0.05 was considered significant; ns = non-significant, * *p* < 0.05, ** *p* < 0.01). (**I**,**J**) Gene expression changes in RAS genes were correlated, using Pearson correlation analysis, with changes in *HIF1A* expression after TMZ + RT treatment across the 12 cell lines. A significant positive correlation with *HIF1A* was only observed for *ATP6AP2* (**I**), *AGTR1* (**J**), and *AGT* (**K**).

## Data Availability

Research data are stored in an institutional repository and will be shared upon request to the corresponding author.

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
