# Peer review of "Upregulation of the Renin–Angiotensin System Is Associated with Patient Survival and the Tumour Microenvironment in Glioblastoma"

_cells, 2024, doi:10.3390/cells13070634_

Round 1

Reviewer 1 Report

Comments and Suggestions for Authors

In this manuscript, the authors delve into the expression of six RAS-related genes (ATP6AP2, AGTR1, AGTR2, ACE, AGT and REN) in TCGA glioblastoma cohort (~ 150 cases) and studied its association with i) survival and ii) tumor microenvironment related pathways as well as the established molecular subtypes of glioblastoma. Specifically, they observed that ATP6AP2, AGTR1, AGTR2 and ACE were highly expressed, while AGT and REN were lowly expressed in glioblastoma compared to LGG. In their multivariate cox regression analysis, significant associations were observed between high expression of AGTR1 and low PFS. Further, the expression of RAS receptors (ATP6AP2, AGTR1 and AGTR2) positively correlated with gene sets related to hypoxia activation, microvasculature, stem cell plasticity and the classical and the mesenchymal glioblastoma subtypes and negatively correlated with the proneural subtype. Finally, the expression of RAS related genes was studied in 12-patient derived glioblastoma cell lines treated with chemoradiation. Interestingly, ATP6AP2, AGTR1 and ACE were upregulated after radiation and temozolomide treatment. Additionally, increase in ATP6AP2 and AGTR1 positively correlated with HIF1A expression changes. Overall, this study examines the relation between expression of RAS genes and patient survival, tumor microenvironment features, and response to the chemoradiation in patient derived cell lines.

Major comments

1. In this study, the authors have focused on the expression of six RAS genes (ATP6AP2, AGTR1, AGTR2, ACE, AGT and REN). Is there a reason for choosing these six particular genes. Why other genes related to RAS pathway like ACE2, CMA1, LNPEP, ENPEP, MME, PRCP, MAS1, etc. were not included. Will it not be beneficial to look at those too?

2. Regarding the ssGSEA analysis, it would be helpful to know how were the TME gene sets identified. Why were only four TME-related gene sets studied.

3. What other genes were upregulated across glioblastoma cell lines after TMZ+RT treatment. Why did the authors focus only on H1F1A. Was it corrected for multiple testing?

Minor comments

1. Please include the references for the below statements:

Lines 41-46

Lines 52-55

2. Fig 1A seems redundant since the same expressions are also provided in panels Fig 1B-G. Also, it would be beneficial to describe what the boxes refer to in the panels. Did the authors see similar trend with the normal tissue?

3. Line 410: AGT is not highly expressed in glioblastoma compared to LGG.

Author Response

Response to reviewer 1

Comment 1

In this study, the authors have focused on the expression of six RAS genes (ATP6AP2AGTR1, AGTR2, ACE, AGT and REN). Is there a reason for choosing these six particular genes. Why other genes related to RAS pathway like ACE2, CMA1, LNPEP, ENPEP, MME, PRCP, MAS1, etc. were not included. Will it not be beneficial to look at those too?

Response to Comment 1

The authors thank the reviewer for this suggestion. The six RAS genes were investigated as they are most widely studied of the RAS genes and are druggable targets. We did further analysis on ACE2 and MAS1 as they represent potential clinical targets for the renin-angiotensin system and play roles in opposing the physiological effect of Ang II. We investigated the baseline expression of ACE2 and MAS1 in the TCGA PanCancer cohort and compared expression of glioblastoma cases versus lower grade glioma and visualised the expression in a new figure within the supplementary material (Figure S2).

The results were communicated within the manuscript (line 204):

“In addition to the six RAS genes, the expression of ACE2 and MAS1 were investigated (Figure S2). ACE2 and MAS1 are known to oppose the physiological effects of elevated RAS activity [37,38]. Both ACE2 and MAS1 were significantly lower in expression in glioblastoma compared to lower grade glioma, while also decreased in glioblastoma compared to other tumour types (Figure S2).”

Comment 2

Regarding the ssGSEA analysis, it would be helpful to know how were the TME gene sets identified. Why were only four TME-related gene sets studied.

Response to Comment 2

The selected TME-related gene sets were used as they represent TME pathways that play substantial roles in tumour aggression and resistance to treatment. The gene sets have been demonstrated to be enriched for such TME pathways, as cited within the results (sentence starting on line 116). However, other TME pathways relating to the immune system and tumour heterogeneity were not investigated as they were outside the scope of this study. This has been acknowledged within the discussion as a future direction (line 536):

“In future work, the link between the RAS and other TME pathways that influence a pro-tumour environment, such as the immune system and tumour heterogeneity [49,50], could also be explored.”

Comment 3

What other genes were upregulated across glioblastoma cell lines after TMZ+RT treatment. Why did the authors focus only on H1F1A. Was it corrected for multiple testing?

Response to Comment 3

Regarding the analysis as shown in Figure 4A-H, this analysis was performed on a number of DNA repair and immune-related genes published within our previous paper (doi.org/10.18632/oncotarget.28551) that first published this RNAseq data. An additional sentence was added to the results section to communicate this (line 336):

“The expression of RAS genes was investigated after the combination of a clinically relevant dose of TMZ (35μM) with RT (2Gy) (Figure 4B-H). Previously, we showed a significant increase in expression of DNA repair and pro-inflammatory related genes after chemoradiation across the 12 glioblastoma cell lines [35].”

HIF1A was focused as this is a master regulator of the transcriptional and cellular response to hypoxia (doi.org/10.1101/gad.12.2.149). As the TCGA data analysis showed a positive correlation between RAS receptor expression and hypoxia, HIF1A expression was analysed in the context of TMZ+RT treatment to examine an association with RAS gene expression in the glioblastoma cell lines. A sentence within the results section was added to emphasis this point (line 344):

“As hypoxia activation was positively correlated with RAS receptor expression (Figure 3B), the association with HIF1A and RAS gene expression was investigated after glioblastoma cell lines were treated with chemoradiation.”

As stated within the figure legend of Figure 4, a paired t-test was performed between the DMSO and TMZ+RT treatment groups across all data points (n = 24) of the 12 cell lines with 2 x biological replicates.

Comment 4

Please include the references for the below statements:

Lines 41-46

Lines 52-55

Response to Comment 4

A reference was added on line 44 (doi.org/10.1038/s41571-020-00447-z) and on line 53 (doi.org/10.1002/cphy.c130040) of the manuscript.

Comment 5

Fig 1A seems redundant since the same expressions are also provided in panels Fig 1B-G. Also, it would be beneficial to describe what the boxes refer to in the panels. Did the authors see similar trend with the normal tissue?

Response to Comment 5

Figure 1B-G illustrate the comparison of RAS gene expression between glioblastoma and lower grade glioma, giving context as to whether these genes are present in higher or lower quantities in the more aggressive and treatment resistant glioblastoma compared to lower grade glioma, while Figure 1A shows the distribution of expression of the RAS genes in just glioblastoma.

Gene expression samples downloaded from the Genomics Data Commons (GDC) Data Portal (https://portal.gdc.cancer.gov/) contained 5 solid tissue normal samples that could be used to potentially compare between RAS gene expression versus glioblastoma. However, this represented a very small sample size of 5 normal tissue samples compared to the > 150 glioblastoma samples, and no matched glioblastoma and normal tissue samples were present in this dataset. Thus, comparison between these two groups may present a bias.

Comment 6

Line 410: AGT is not highly expressed in glioblastoma compared to LGG.

Response to Comment 6

This error was corrected on line 489:

“The majority of RAS genes (ATP6AP2, AGTR1, AGTR2 and ACE) in our study were…”

Reviewer 2 Report

Comments and Suggestions for Authors

"Glioblastoma stands out as an exceptionally aggressive disease with dismal survival rates. The renin-angiotensinogen system (RAS), pivotal in blood pressure regulation, emerges as a crucial player in glioblastoma pathogenesis, potentially fostering tumor progression and influencing patient outcomes. This study delves into the intricate landscape of RAS gene expression, including ATP6AP2, AGTR1, AGTR2, ACE, AGT, and REN, utilizing glioblastoma patient samples sourced from The Cancer Genome Atlas (TCGA). By scrutinizing their correlation with survival metrics and the expression of tumor microenvironment (TME) pathways, the researchers shed light on the intricate interplay between RAS components and disease progression. Furthermore, they meticulously examine alterations in RAS gene expression following chemoradiation treatment across 12 patient-derived glioblastoma cell lines. The investigation extends beyond glioblastoma, offering a comprehensive comparative analysis of RAS gene expression across TCGA Pan Cancer tumors.

I have several comments:

1.    The provided information on the non-significant trend in survival outcomes for REN-altered groups in glioblastoma was not found in Figure S2 as indicated. The authors might need to review the source of this data for clarity. “In glioblastoma, however, a non-significant trend was observed for PFS (HR = 1.33 (0.94-1.9), p = 0.11) and OS (HR = 1.37 (0.94-1.98), p = 0.1) where the REN altered group (n = 36) had shorter PFS (6.7 months) and OS (11.1 months) compared to the non-altered group (n = 542) (PFS: 7.2 months , OS: 14.5 months). “

2.    To strengthen the understanding of the relationship between RAS receptor expression and tumor microenvironment pathways, the authors could consider conducting siRNA or shRNA experiments targeting multiple RAS genes. This would help confirm whether the knockdown of these genes indeed leads to differences before and after chemoradiation, providing valuable insights into the functional relevance of RAS expression in glioblastoma."

Author Response

Response to reviewer 2

Comment 1

The provided information on the non-significant trend in survival outcomes for REN-altered groups in glioblastoma was not found in Figure S2 as indicated. The authors might need to review the source of this data for clarity. “In glioblastoma, however, a non-significant trend was observed for PFS (HR = 1.33 (0.94-1.9), = 0.11) and OS (HR = 1.37 (0.94-1.98), = 0.1) where the REN altered group (n = 36) had shorter PFS (6.7 months) and OS (11.1 months) compared to the non-altered group (n = 542) (PFS: 7.2 months , OS: 14.5 months).

Response to Comment 1

To address this, we have added Kaplan-Meier survival plots of the glioblastoma cohort with REN CNA alterations vs non-alterations in Figure S3 (previously Figure S2) for PFS (D) and OS (E) and displayed the univariate cox regression results (hazard ratio and p value).

Comment 2

To strengthen the understanding of the relationship between RAS receptor expression and tumor microenvironment pathways, the authors could consider conducting siRNA or shRNA experiments targeting multiple RAS genes. This would help confirm whether the knockdown of these genes indeed leads to differences before and after chemoradiation, providing valuable insights into the functional relevance of RAS expression in glioblastoma.

Response to Comment 2

The authors acknowledge and thank the author for this suggestion. This is a very valid future direction and thus an additional sentence within the discussion of the limitations to the study has been added (line 587):

“In future work, the use of siRNA or shRNA to knockdown such RAS genes in combination with chemoradiation may also give insight into the feasibility of targeting the RAS, as well as validate the observations of increased RAS gene expression in the glioblastoma cell lines. “